# Magnetic, electronic, and structural investigation of the strongly correlated $Y_{1-x}Sm_xCo_5$ system

F. Passos[1], D. Cornejo[2], G. J. Nilsen[3], V. Martelli[1] and J. Larrea Jiménez[1]⋆

**1** Laboratory for Quantum Matter under Extreme Conditions, Institute of Physics, University of São Paulo, São Paulo, Brazil
**2** Institute of Physics, University of São Paulo, São Paulo, Brazil
**3** ISIS Neutron and Muon Source, Rutherford Appleton Laboratory, Didcot OX11 0QX, United Kingdom

⋆ larrea@if.usp.br

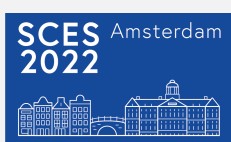

## Abstract

$SmCo_5$ and $YCo_5$ are isostructural compounds both showing large magnetocrystalline anisotropy, where the former originates mainly from the crystal-electric field and magnetic interactions. We investigate the contribution of both interactions by partially substituting Y by Sm in as-cast polycrystalline $Y_{1-x}Sm_xCo_5$ (with $x=$0.00, 0.12, 0.18, 0.31, and 0.40) and measuring their structural, magnetic, and electrical properties through X-ray diffraction, magnetization, and electrical transport measurements. Our results suggest an interplay between microstructure strain in as-cast samples and the electronic and magnetic interactions.

## 1 Introduction

$SmCo_5$ alloys are well-known high-performance rare-earth permanent magnets which have been intensively studied in the last five decades due to their high magnetization saturation, high Curie temperature, and high coercivity [1]. $SmCo_5$ has a hexagonal $CaCu_5$-type crystal structure (space group P6/mmm) and its excellent magnetic properties result from the large magnetocrystalline anisotropy originated mainly from the Sm localized $4f$-electrons interaction with the hexagonal crystal field (CF) [2,3]. On the other hand, the isostructural compound $YCo_5$ also has a considerably high magnetocrystalline anisotropy but it originates from the spin-orbit coupling of the large orbital angular momentum of the Co $3d$-electrons [3,4].

$Y_{1-x}Sm_xCo_5$ is a promising candidate system to understand the role of the rare-earth localized $f$-electrons and the contribution of the transition metal itinerant $d$-electrons to the

Table 1: Physical properties and parameters obtained from the magnetization and electrical transport data for each $Y_{1-x}Sm_xCo_5$ sample, where $\sigma_S$ is the saturation magnetization, $H_C$ is the coercivity, *RRR* is the residual-resistance ratio, and $\Delta$ is the magnon gap.

| $x$ | $\sigma_S$ (emu/g) | $H_C$ (Oe) | *RRR* | $\Delta$ (K) |
|---|---|---|---|---|
| 0.00 | 88(3) | 203(8) | 1.38(1) | 40(5) |
| 0.12(1) | 70(3) | 117(6) | 1.35(1) | 53(7) |
| 0.18(1) | 68(3) | 123(6) | 1.31(1) | 46(6) |
| 0.31(2) | 69(3) | 153(6) | 1.36(1) | 40(6) |
| 0.40(1) | 53(5) | 218(12) | 1.32(1) | 16(7) |

magnetocrystalline anisotropy and magnetic properties. Moreover, the different atomic ratios of Sm and Y allow to investigate the effect of changing the atomic spacing in the packed hexagonal crystal structure on the physical properties [4]. We have successfully synthesized polycrystalline $Y_{1-x}Sm_xCo_5$ ($x$=0.00, 0.12, 0.18, 0.31, and 0.40) solid solutions and investigated, for the first time, the influence of the partial substitution of Y by Sm in the structural, magnetic, and electrical properties.

## 2   Experimental details

Polycrystalline $Y_{1-x}Sm_xCo_5$ samples with $x$=0, 0.12, 0.18, 0.31, and 0.4 were synthesized by arc-melting the stoichiometric amounts of high-purity parent materials under an Argon atmosphere. The X-ray diffraction (XRD) of the samples was performed with the Cu-$K_\alpha$ radiation using a Rigaku Ultima III diffractometer with a monochromator. Scanning electron microscopy (SEM) and energy-dispersive X-ray spectroscopy (EDS) were used to determine the stoichiometry in each sample.

The magnetization vs magnetic field (M-H) hysteresis of each sample was measured using a vibrating-sample magnetometer (VSM) at room temperature with a maximum magnetic field of 15 kOe. For the electrical transport measurements, the samples were polished to acquire a regular shape and the electrical contacts on the samples were made using silver conducting paint and 50 $\mu$m gold wires. The resistance of the samples was measured using the standard four-probe method at temperatures from 10 K to 290 K. Table 1 shows the Sm concentration $x$ obtained by EDS with the respective error bar and some of the physical properties of the samples discussed in the following section.

## 3   Results and discussion

### 3.1   Structural properties

Figure 1 shows the XRD pattern of the as-cast $Y_{1-x}Sm_xCo_5$ ($x$=0.00, 0.12, 0.18, 0.31, and 0.40) samples and their respective Rietveld refinements. We can successfully recognized the diffraction peaks associated with the $CaCu_5$-type hexagonal crystal structure, besides a small $Y_2O_3$ phase (see Fig. 1). Fig. 2 shows the lattice parameters and the relative volume change as function of Sm substitution $x$, obtained by the Rietveld refinement of our XRD patterns.

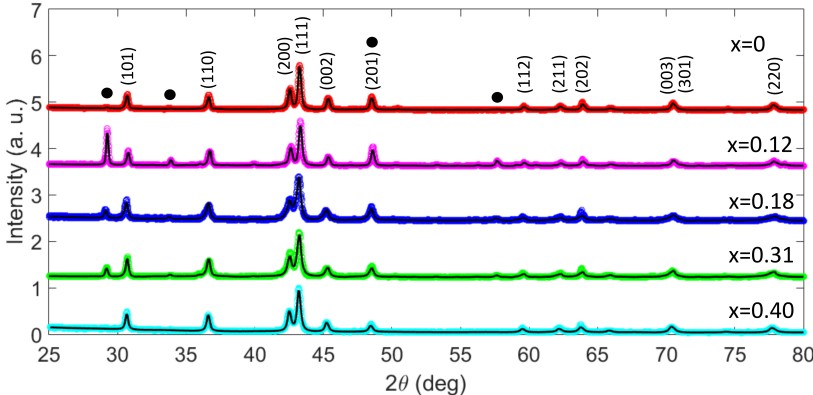

Figure 1: XRD patterns (circles) and Rietveld refinements (solid lines) of the poly-crystalline $Y_{1-x}Sm_xCo_5$ samples (with x=0.00, 0.12, 0.18, 0.31 and 0.40). The Miller indexs correspond to the main peaks of the $YCo_5$ phase and the black dots indicate the peaks from the $Y_2O_3$ phase.

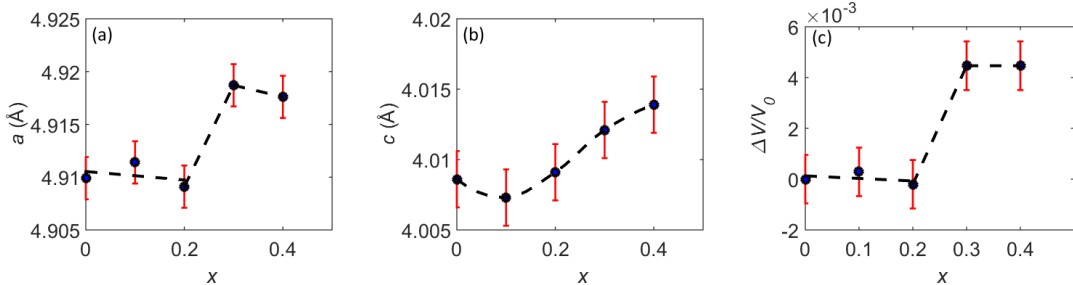

Figure 2: Lattice parameters $a$ and $c$, and unit cell volume of the $Y_{1-x}Sm_xCo_5$ samples for different Sm doping $x$ obtained from the Rietveld refinement analysis of our X-ray diffraction patterns.

We observe that the lattice parameter $c$ (Fig. 2b) monotonically increases with Sm substitution $x$, except for $x$<0.12. On the other hand, the lattice parameter $a$ (Fig. 2a) does not show the linear variation expected from the Vergard's law, remaining almost constant up to $x \approx 0.2$, with a subsequent sizable rapid increase around $x$=0.3. This change in the lattice between $x$=0.18 and $x$=0.31 becomes noticeable also in the variation of volume as a function of doping (see Fig. 2c), indicating a structural deformation.

## 3.2 Magnetic properties

Figure 3 shows the M-H hysteresis loops of $Y_{1-x}Sm_xCo_5$ for each Sm substitution. The hysteresis loops suggest a ferromagnetic order for all compositions.

From the hysteresis loops we obtain the coercivity ($H_C$) and the saturation magnetization ($\sigma_S$) for each sample. Fig. 4 shows the evolution of these properties with Sm substitution.

We note that the saturation magnetization shows a linear decrease with the Sm substitution. On the other hand, we observe that the coercivity first slightly decreases from $x$=0 to $x$=0.12, and then starts to increase monotonically with the increasing of Sm substitution, in a very similar way to what we observed in the $c$ lattice parameter. This similar dependence of the coercivity and the $c$ lattice parameter suggests magneto-elastic coupling effects [5,6] and can also be related to an interplay between the change on the hexagonal crystal structure when substituting Y by Sm and the (Y,Sm)$Co_5$ large uniaxial magnetocrystalline anisotropy along the $c$-axis [4], revealing as the easy-axis magnetization.

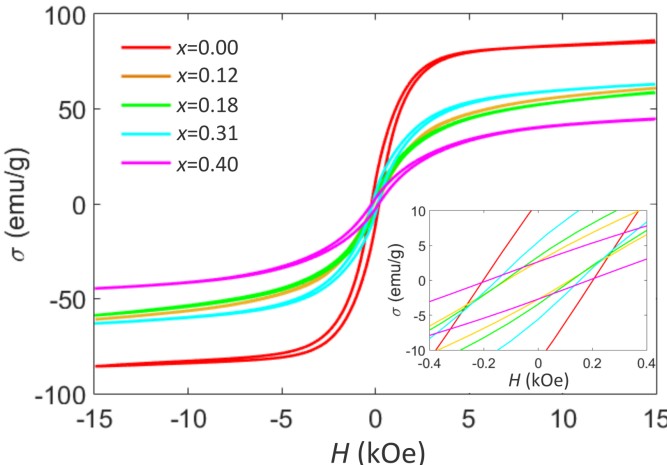

Figure 3: Bulk magnetization as a function of the magnetic field of the $Y_{1-x}Sm_xCo_5$ samples with different dopings of Sm measured at room temperature. The inset shows a zoom to better visualize the coercivity of each sample.

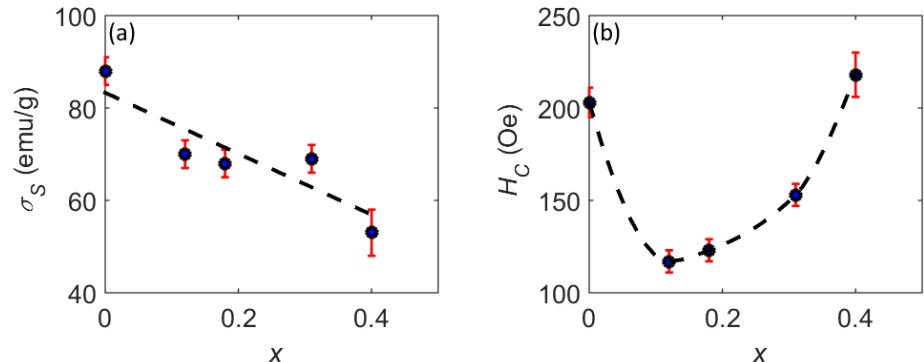

Figure 4: (a) Saturation magnetization and (b) coercivity as a function of Sm doping in the $Y_{1-x}Sm_xCo_5$ samples. Dashed lines are guide to eyes.

### 3.3 Electrical transport measurements

We compare the temperature dependence of the electrical resistance normalized to its room temperature value for all the five samples, as shown in Fig. 5. We observe that for all samples the resistance shows a decrease with temperature very similar to the observed in other metals [7].

We can separate the contributions to the total resistivity as follows:

$$\rho(T) = \rho_0 + \rho_{ee}(T) + \rho_{mag}(T) + \rho_{ph}(T), \tag{1}$$

where $\rho_0$ is the residual resistivity, $\rho_{ee}$ is the contribution from the electron-electron interaction, $\rho_{mag}$ is the magnetic contribution from the electron-spin wave scattering, and $\rho_{ph}$ is the contribution coming from the electron-phonon scattering processes. The dominant scattering mechanism varies significantly among materials and with the temperature range. In the case of metals at low temperatures, the strongly interacting electron-electron scattering is usually well described by the Fermi liquid model which results in a $T^2$ dependence of the resistivity. On the other hand, for ferromagnets, the magnon modes quadratic dispersion relation $\omega(k) = \Delta + Dk^2$ gives rise to a magnetic contribution to the resistivity with the following

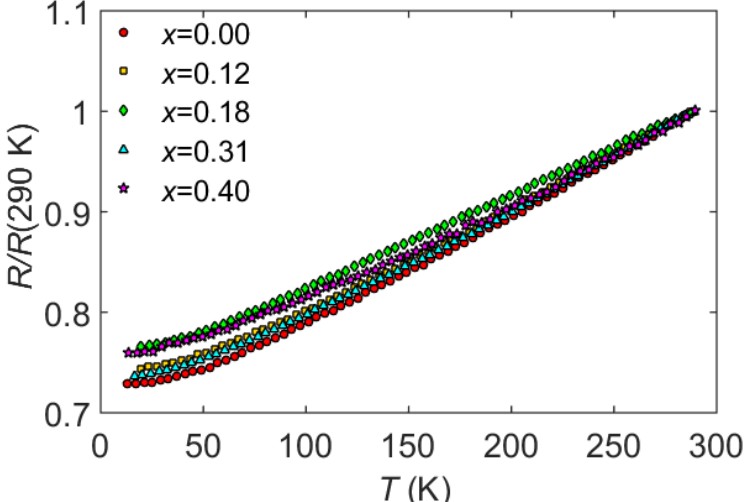

Figure 5: Electrical resistances normalized at 290 K for polycrystalline $Y_{1-x}Sm_xCo_5$.

temperature dependence for $k_B T \ll \Delta$ [8]:

$$\rho_{mag}(T) = b\Delta T e^{-\Delta/T}\left(1 + 2\frac{T}{\Delta}\right), \qquad (2)$$

where $\Delta$ is the spin-wave gap, $D$ is the spin-wave stiffness, and $b$ is a constant such that $b \propto 1/D^2$. Finally, the electron-phonon interactions usually has a $\rho \propto T^5$ contribution at lower temperatures, and a $\rho \propto T$ contribution at higher temperatures [9, 10].

The attempt to include the $T^5$ contribution from the phonons in the fittings at lower temperatures (below 60 K) did not pay off, which indicates that phonon scattering is not significant. Indeed, it has already been suggested that the phonon contribution for $RCo_5$ systems in this temperature range is much smaller compared to the other contributions [9] and, therefore, can be neglected. Furthermore, for the $YCo_5$ sample, a quadratic dependence of the resistivity up to 60 K was already observed previously [10]. Therefore, we perform a fitting of the electrical resistivity below 50 K for each sample including only the Fermi liquid and magnetic contributions using the following equation:

$$\rho(T) = \rho_0 + AT^2 + b\Delta T\left(1 + \frac{2T}{\Delta}\right)e^{-\Delta/T}, \qquad (3)$$

where $A$ is the quadratic parameter from the electron-electron scattering. Fig. 6 shows that the fittings of eq. 3 have a good agreement with the experimental data up to 50 K and Fig. 7 shows the values of all parameters obtained from the fittings of the resistivity data as a function of the Sm substitution.

We can note in Fig. 7a that the residual-resistance ratio (RRR) initially decreases almost linearly until $x$=0.18 followed by an increase at $x$=0.31. Besides that, the $A$ and $b$ parameters (Figs. 7b and c) show an overall tendency of decrease except at $x$=0.31 where all of them present an abrupt increase. This abrupt change agrees with the in-plane lattice parameters (see Fig. 2a) and unit cell volume (see Fig. 2c) changes at $x$=0.31, which suggests strain effects on the electronic and magnetic scattering mechanisms [6]. The decrease of the $b$ parameter (Fig. 7c) means an increase of the spin-wave stiffness $D$ ($b \propto 1/D^2$) and, therefore, a hardening of the magnon modes with increasing Sm substitution. Finally, in Fig. 7d we observe that the magnon gap has a maximum around $x$=0.12 and then starts to decrease monotonically, which might indicate a magnetic anisotropy loss.

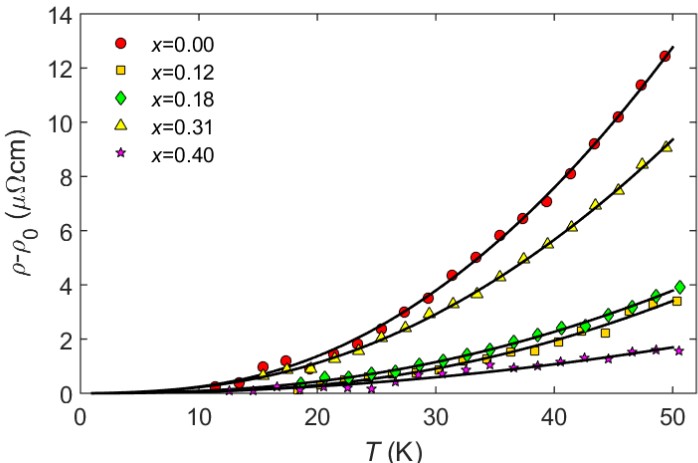

Figure 6: Temperature dependence of the difference between the electrical resistivity and the residual electrical resistivity ($\rho_0$) for our polycrystalline samples $Y_{1-x}Sm_xCo_5$. Solid lines are fittings according to eq. 3.

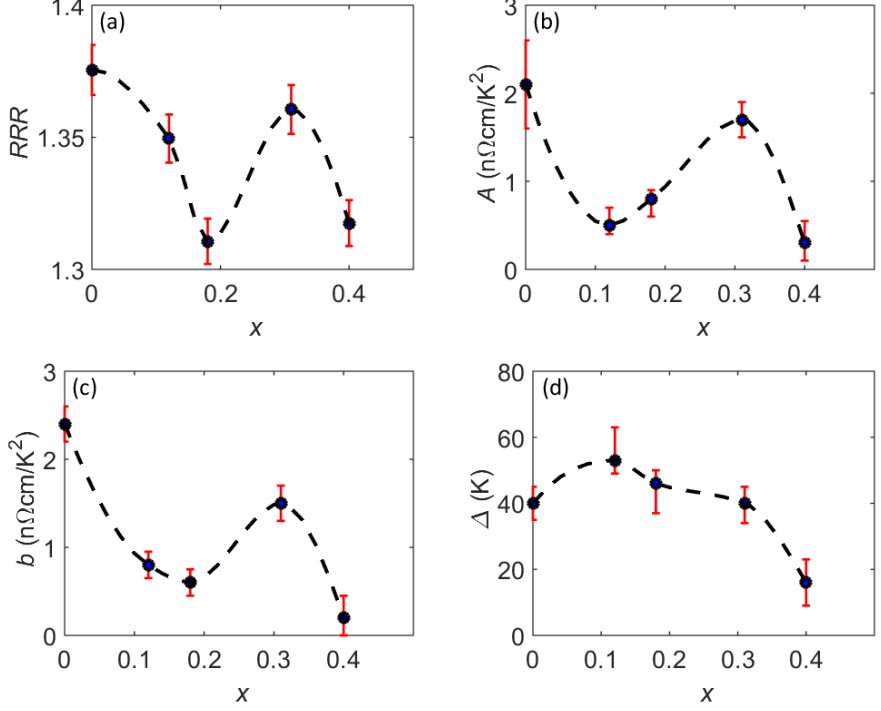

Figure 7: Parameters obtained fitting the experimental data with equation 3 for each $Y_{1-x}Sm_xCo_5$ sample. (a) Residual-resistance ratio (RRR) obtained from $\rho_0$ using RRR=$\rho(290\ K)/\rho_0$. (b) Quadratic parameter from the Fermi liquid contribution. (c) Magnetic contribution parameter where $b \propto 1/D^2$. (d) Spin-wave gap from the magnetic contribution.

## 4    Conclusion

Polycrystalline $Y_{1-x}Sm_xCo_5$ as-cast samples were successfully synthesized, where the physical properties were not dramatically affected by the presence of a small spurious yttrium/samarium oxide phase. Our results indicate an abrupt change in the lattice parameter $a$

and in the unit cell volume (Fig. 2) at the same Sm substitution of $x=0.31$, which influences the electric and magnetic scattering process. In addition, similar dependence of lattice parameter $c$ and the coercivity with the Sm substitution suggests the important role that the $f$-electron orbitals plays on the magnetocrystalline anisotropy and crystal electrical field, both underlying mechanisms to understand the magnetic ground state in $RCo_5$ magnets. Finally, the electrical transport measurements indicated the presence of both electron-spin and electron-electron interactions and an apparent hardening of the magnon modes with Sm substitution.

Our results trigger further investigation at higher substitutions of Sm in order to understand the role that strong correlation between itinerant and localized electrons plays on setting the magnetic ground state in hard magnets.

# Acknowledgements

We acknowledge M.C.A. Fantini for the access to the Laboratory of Crystallography (LCr-IFUSP).

**Funding information** JLJ acknowledges support from JP-FAPESP (2018/08845-3) grant and CNPq-PQ2 (310065/2021-6) fellowship. FP acknowledges FAPESP 2019/24711-0 grant. VM acknowledges JP-FAPESP (2018/19420-3) grant and GN acknowledges FAPESP 2019/24797-1 grants.

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
