# Peer review of "Magnetic, electronic, and structural investigation of the strongly correlated Y$_{1-x}$Sm$_{x}$Co$_{5}$ system"

_SciPost Physics Proceedings, doi:SciPost Phys. Proc. 11, 021 (2023)_

## Round 1 · Referee Report · Anonymous (Referee 1) · 2022-11-2

Strengths

1.) The manuscript describes the influcence of Sm-doping on the structural, magnetic, and electrical properties of polycrystalline Y1−x Smx Co5 (with x=0, 0.1, 0.2, 0.3, and 0.4). The main result is the interplay between microstructure strain of the different samples and their electronic and magnetic interactions.

2.) The paper is clearly written and presents new and important results of this class of magnetic materials.

Weaknesses

1.) I recommend to comment on the error bars of the Sm-concentration x for the polycrytalline samples or add a reference for typical variations in the Sm- concentrations for related systems.

2.) I strongly recomment to add the transport data for x = 0.1 and x = 0.4 and the fits according eq. 3, since their fitting parameters are presented in fig. 7

Report

As stated above the paper is clearly written and deserves publication in
the proceedings of the SCES conference. However, I strongly
recommend to show the transport data for x =0.1 and x = 0.4 in figure 6.

Requested changes

1.) comment on the error bars for the Sm concentration (see above)
2.) add the data for x = 0.1 and x = 0.4 in fig. 6 (see above)

  • validity: good
  • significance: good
  • originality: high
  • clarity: high
  • formatting: good
  • grammar: good

Author:  Fernando Passos  on 2023-01-23  [id 3262]

(in reply to Report 1 on 2022-11-02)

We thank the referee for his/her work revising our manuscript, where he/she judges our manuscript deserves a publication in the conference proceedings after minor corrections. We performed EDS measurements on our samples to determine the stoichiometry with the respective error bars. We are resubmitting our manuscript including all the corrections the referee pointed out.

---

## Round 2 · Author Response

We thank the referee for his/her work revising our manuscript, where he/she judges our manuscript deserves a publication in the conference proceedings after minor corrections. We performed EDS measurements on our samples to determine the stoichiometries with the respective error bars. We we are resubmitting our manuscript including all the corrections the referee pointed out.

---

## Round 2 · List of Changes

1) We included in the text and Table 1 the standard deviation in the Sm composition according to SEM and EDS analysis.

2) We included the data for x=0.12 and x=0.4 in Figure 6.

---

## Editorial Decision

published